# Stem Extract from *Momordica cochinchinensis* Induces Apoptosis in Chemoresistant Human Prostate Cancer Cells (PC-3)

**DOI:** 10.3390/molecules27041313

**Published:** 2022-02-15

**Authors:** Seksom Chainumnim, Audchara Saenkham, Kulvadee Dolsophon, Kittipong Chainok, Sunit Suksamrarn, Wanlaya Tanechpongtamb

**Affiliations:** 1Department of Biochemistry, Faculty of Medicine, Srinakharinwirot University, Bangkok 10110, Thailand; seksom.chainumnim@g.swu.ac.th; 2Department of Chemistry and Center of Excellence for Innovation in Chemistry, Faculty of Science, Srinakharinwirot University, Bangkok 10110, Thailand; audchara.sk@gmail.com (A.S.); kulvadee@g.swu.ac.th (K.D.); 3Thammasat University Research Unit in Multifunctional Crystalline Materials and Applications (TU-MCMA), Faculty of Science and Technology, Thammasat University, Khlong Luang, Pathum Thani 12121, Thailand; kc@tu.ac.th

**Keywords:** *Momordica cochinchinensis*, stem extract, apoptosis, chemoresistant, prostate cancer, Noxa

## Abstract

Natural compounds have been recognized as valuable sources for anticancer drug development. In this work, different parts from *Momordica cochinchinensis* Spreng were selected to perform cytotoxic screening against human prostate cancer (PC-3) cells. Chromatographic separation and purification were performed for the main constituents of the most effective extract. The content of the fatty acids was determined by Gas Chromatography-Flame Ionization Detector (GC–FID). Chemical structural elucidation was performed by spectroscopic means. For the mechanism of the apoptotic induction of the most effective extract, the characteristics were evaluated by Hoechst 33342 staining, sub-G1 peak analysis, JC-1 staining, and Western blotting. As a result, extracts from different parts of *M. cochinchinensis* significantly inhibited cancer cell viability. The most effective stem extract induced apoptosis in PC-3 cells by causing nuclear fragmentation, increasing the sub-G1 peak, and changing the mitochondrial membrane potential. Additionally, the stem extract increased the pro-apoptotic (caspase-3 and Noxa) mediators while decreasing the anti-apoptotic (Bcl-xL and Mcl-1) mediators. The main constituents of the stem extract are α-spinasterol and ligballinol, as well as some fatty acids. Our results demonstrated that the stem extract of *M. cochinchinensis* has cytotoxic and apoptotic effects in PC-3 cells. These results provide basic knowledge for developing antiproliferative agents for prostate cancer in the future.

## 1. Introduction

Prostate cancer is the most frequently diagnosed cancer in men worldwide [1]. In 2020, 191,930 (21%) new cases and 33,330 (10%) prostate-cancer-related deaths were reported. The incidence of prostate cancer increases with age and it is more likely to develop at age 65 or older [1]. Additionally, its incidence tends to increase steadily in both developed and developing countries. Most prostate cancer patients usually progress to the metastatic and chemoresistance stages, which are obstacles to treatment. Thus, searching for alternative treatments or better chemotherapeutic drugs is significant for patient survival.

To date, several targets for anticancer drug development have been identified. One target mechanism that is often used during drug investigation is apoptosis or programmed cell death. This is a normal process that occurs under various physiological or pathological situations in our body. It is involved in normal development, function, and homeostasis, especially for controlling the number of cells [2]. The initiation of apoptosis can be triggered by diverse extracellular and intracellular factors, such as harmful carcinogens or mutagenic agents, viral infections, ultraviolet radiation, growth factor withdrawal, and inflammation. The characteristics of apoptotic cells are highly specific, with DNA fragmentation, cell shrinkage, and membrane blebbing. To induce apoptosis, two major pathways, intrinsic and extrinsic, are well recognized. The extrinsic pathway is triggered by the interaction of cell death receptors with their ligands, while the intrinsic pathway is related to mitochondrial dysfunction. Many protein mediators are involved in these two pathways, which can be grouped as pro-apoptotic (Bax, Bak, Bad, Bid, Bim, Puma, Noxa, etc.) and anti-apoptotic (Bcl-2, Bcl-XL, Mcl-1, etc.) molecules [3]. The activation of the caspase cascade is the main mediator of both pathways and it is well accepted as a specific marker of apoptotic signaling.

Although most anticancer drugs that are currently used in the clinic have the ability to induce apoptosis in many cancers, there are some aggressive or resistant cancer types that cannot be successfully cured with either synthetic or natural agents. Hence, new anticancer drugs must be developed. *Momordica cochinchinensis* (Lour.) Spreng (MC) or “Gac” (belonging to the Cucurbitaceae family) is a type of perennial vine grown throughout northeastern Australia and southeastern Asian countries, including Thailand [4]. MC fruit has been commonly used, mainly as food and traditional medicine for various diseases such as cardiovascular disease and arthritis [5].

Different parts of MC were extracted to identify the various candidate compounds. MC arils have been found to contain a number of phytochemicals, such as fatty acids, phenolics, flavonoids, and carotenoids (mostly lycopenes and beta-carotenes). The quantities of both lycopene and beta-carotene in MC fruit are reported to be much higher than those in other plants [6]. Additionally, numerous studies have shown that these phytochemicals possess interesting biological effects, including cardio- and testicular protective effects [7,8], anticancer effects [9,10,11,12], and antioxidant and anti-inflammatory effects [13]. Meanwhile, MC seeds contain saponins, saponin glycosides, and macrocyclic peptides (e.g., momordins I-III, momordins Ia-c, and momordins IIa-d), which also exhibit noticeable activities, such as anticancer [14,15,16,17], antioxidant [18], anti-inflammatory [19,20], and reno-protective activities [21]. However, the chemical composition and biological activity of other parts of MC, including its leaves, seeds, stems, and roots, have not yet been elucidated. Therefore, the present work aimed to explore the anticancer activity of these parts of MC and investigate the possibility of the active extract having apoptosis-inducing activity. In addition, the main constituents of this active extract were chemically characterized.

## 2. Results and Discussion

### 2.1. The Stem Extract Demonstrated Antiproliferative Effects against Cancer Cells but Not against Normal Cells

To study the antiproliferative activity of the MC extract against cancer cells, several types of cancer cell models were selected for screening, including A549, HeLa, MDA-MB-231, and PC-3 cells. To prepare the MC crude extracts, different parts of MC (leaves, seeds, stems, and roots) were extracted with methanol following standard procedures. The antiproliferative activity of each soluble fraction against cancer cells was defined using an MTT assay. The cells were treated with or without several concentrations of MC extracts at 0–5 mg/mL for 24 h, while 0.5% DMSO was used as the vehicle control. The results were demonstrated as shown in Table 1. All extracts exhibited antiproliferative effects in all cancer cells tested, but at different levels. Among these crude extracts, the stem extract showed the highest efficiency in all four types of cancer cells. The IC_50_ values for A549, HeLa, MDA-MB-231, and PC-3 cells were 0.44, 0.42, 0.62, and 0.62 mg/mL, respectively. These results are consistent with other studies. The antiproliferative activity against cancer cells of a water extract from aril parts was in the range of 0.49–0.73 mg/mL [10]. Thus, the stem extract was selected for the following experiments. One important characteristic of an ideal anticancer agent is that it is less toxic to normal cells, so Vero was used as a representative of normal cells. For a model of cancer cells, PC-3 was selected for investigation due to its chemoresistance and high metastasis potential. As illustrated in Figure 1, the IC_50_ values of the PC-3 and Vero cells were 0.62 and 2 mg/mL, respectively. These data revealed that the stem extract exhibited more specific inhibitory activity against cancer cells. Thus, the mechanism of cancer cell death induction by the stem extract was evaluated along with a characterization of its chemical composition.

### 2.2. Stem-Extract-Induced Apoptosis in PC-3 Cells

As shown in the cytotoxic screening, the stem extract affected all of the cancer cells tested, so we investigated the potential mechanism by which the stem extract induces cancer cell death. Apoptosis was selected as a target because it is the main pathway triggered by most anticancer agents [2,22]. To confirm this process, morphological changes, such as chromatin condensation, apoptotic bodies, membrane blebbing, and cell shrinkage, were evaluated. PC-3 cells were treated with or without stem extract at 0, 0.3, 0.6, 0.9, and 1.2 mg/mL for 24 h, followed by Hoechst 33342 staining and the cells were imaged under a fluorescence microscope. Most treated cells exhibited chromatin condensation, and some cells demonstrated nuclear fragmentation (Figure 2). This specific characteristic of apoptotic cells was confirmed by using sub-G1 peak analysis. The method used propidium iodide, a DNA intercalating agent, to stain the population of apoptotic cells that had diminished DNA content [23]. As shown in Figure 3, PC-3 cells were treated with various concentrations of stem extract (0, 0.6, 0.9, and 1.2 mg/mL) exhibited a higher number of sub-G1 populations while increasing the concentration of stem extract: 0.86%, 5.9%, 25%, and 37%, respectively. These results support our hypothesis that stem extract has the ability to induce apoptosis in PC-3 cells.

In addition, the apoptosis process usually causes mitochondrial damage, so we explored whether this effect would also be induced by stem extract. The experiment was conducted by using JC-1 dye staining. It is a fluorescent lipophilic cationic dye that is sensitive to the mitochondrial membrane potential. In normal cells, a high level of mitochondrial membrane potential causes JC-1 to accumulate in mitochondria and it exists in the J-aggregate form, which is shown in the red spectrum. In contrast, JC-1 is present as a monomer in cells with lower potential or mitochondrial damage and yields green fluorescence [24,25]. Therefore, the changes in the red to green ratio could indicate the status between healthy and apoptotic cells. As shown in our results, the stem-extract-treated cells demonstrated a decrease in the red/green ratio in a dose-dependent manner (Figure 4). The percentage of cells with JC-1 green fluorescence was not significantly altered in the control group (3.20%), while in the treated samples, the green fluorescence significantly increased by 8.19%, 8.90%, 22.34%, and 47.24% at 0.3, 0.6, 0.9, and 1.2 mg/mL of stem extract, respectively. The present findings suggested that stem extract could trigger apoptosis in PC-3 cells by inducing morphological changes, increasing the sub-G1 population, and reducing the mitochondrial membrane potential.

### 2.3. The Stem Extract Affected the Expression of Apoptosis-Related Proteins in PC-3 Cells

To deeply investigate the effect of the stem extract on the protein machinery involved in the apoptosis mechanism, several mediators were determined by Western blotting, including procaspase-3, Mcl-1, Noxa, and Bcl-xL. As shown in Figure 5, PC-3 cells treated with stem extract at 0, 0.3, 0.6, 0.9, and 1.2 mg/mL demonstrated a reduction in procaspase-3, Bcl-xL, and Mcl-1 protein expression. The results for these three proteins exhibited the same trend, in which their expression started decreasing from 0.6 mg/mL. Meanwhile, the level of Noxa increased significantly from 0.3–0.9 mg/mL and dropped at 1.2 mg/mL. Caspase-3 is the main executioner caspase in the apoptosis mechanism. Normally, all caspases are present in normal cells as inactive zymogens (pro-form) and they undergo activation into a cleaved form when the apoptosis mechanism is induced. In this case, a reduction in a pro-form could therefore indicate the activation of the apoptosis pathway. Both Bcl-xL and Mcl-1 are members of the Bcl-2 family of proteins that play important roles in apoptosis inhibition. They prevent the release of cytochrome c into the cytoplasm by sequestering Bak and Bax at the mitochondrial membrane [26,27]. Therefore, the stem extract not only decreased anti-apoptotic proteins, but also increased pro-apoptotic proteins. Notably, both Bcl-xL and Mcl-1 are highly expressed in PC-3 cells and are related to the chemotherapeutic resistance of prostate cancer cells. The stem extract is therefore a possible therapeutic agent for prostate cancer treatment. For Noxa, a BH-3-only protein that normally functions as a pro-apoptotic molecule, we found an increase after treatment with the stem extract. Noxa has been indicated to react specifically with Mcl-1 and promote Mcl-1 degradation [28,29,30]. In the present study, a reduction in Mcl-1 may result from an increase in Noxa, which causes Mcl-1 degradation. Thus, the data indicate another possible role of the stem extract that could reverse the chemoresistance property of prostate cancer cells. However, it is noticeable that the amount of stem extract at 1.2 mg/mL or higher may have a strong toxic effect on the cells, as the amount of Noxa was decreased. Hence, the effective concentration of stem extract should be in the range of 0.6–0.9 mg/mL.

### 2.4. Chemical Compositions of the Stem Extract and Its Active Compounds

Since an anticancer activity of the stem extract was clearly demonstrated in PC-3 cells, we asked what active compound could be responsible for this function. Bioassay-guided fractionation of this MC stem extract was then carried out and it yielded two fractions, EtOAc (IC_50_ 1.25 mg/mL) and *n*-BuOH (IC_50_ 1.46 mg/mL). Subsequently, chromatographic phytochemical investigations of the more active EtOAc soluble part produced a pale-yellow oil at 0.16% (*w*/*w*) from fraction E1 as the most abundant fraction, which was quantified for fatty acid content using GC-FID. The percentage of the identified fatty acid composition is presented in Table 2, and the GC-FID chromatogram is shown in Figure 6. The unsaturated fatty acids (USFAs) were characterized as palmitoleic acid (1.34%), oleic acid (7.30%), linoleic acid (11.85%), and *α*-linolenic acid (27.0%), while those of the saturated fatty acids (SFAs) were lauric acid (0.18%), myristic acid (1.54%), palmitic acid (35.07%), stearic acid (14.09%), and arachidic acid (1.63%). Both palmitic acid and *α*-linolenic acid were obviously the main fatty acid constituents obtained from the stem part. Fraction E2-5 and E6.5.4 were collected and proved to be *α*-spinasterol (**1**) and (−) ligballinol (**2**), respectively, by spectroscopic examination, mainly ^1^H, ^13^C NMR, and MS data, as well as X-ray crystallography (Figure 7 and Figure 8) (Appendix A). A comparison of their physicochemical and spectral data with published values was also performed. Compound **2** exhibited a negative specific rotation ${\left[\alpha \right]}_{D}^{25.5}$ −11.5 (c = 0.20, MeOH), and its X-ray crystal structure supported the stereochemical assignment, as shown (Figure 8). Therefore, Compound **2** was deduced to be (−) 4,4′-((1*R*,3a*S*,4*R*,6a*S*)-tetrahydro-1*H*,3*H*-furo[3,4-c] furan-1,4-diyl)diphenol or (−) ligballinol. This compound has also been reported from the MC seed [31,32].

Compound **1** was obtained as a white amorphous powder, mp. 168–169 °C (lit [33] 169 °C). ^1^H-NMR (300 MHz, CDCl_3_) δ_H_: 5.16 (m, 2H, H-7 and H-22), 5.03 (dd, *J* = 15.1, 8.6 Hz, 1H, H-23), 3.59 (m, 1H, H-3), 2.00–0.95 (m, for methine and methylene protons), 1.03 (d, *J* = 6.6 Hz, CH_3_-21), 0.84 (d, *J* = 6.6 Hz, 3H, CH_3_-26), 0.80 (s, 3H, CH_3_-19), 0.79 (d, *J* = 5.5 Hz, 3H, CH_3_-27), 0.55 (s, 3H, CH_3_-18); ^13^C-NMR (75 MHz, CDCl_3_) δ_C_: 139.5 (C-8), 138.1 (C-22), 129.5 (C-23), 117.4 (C-7), 71.0 (C-3), 55.9 (C-17), 55.1 (C-14), 51.2 (C-24), 49.4 (C-9), 43.3 (C-13), 40.7 (C-20), 40.2 (C-5), 39.4 (C-12), 38.0 (C-4), 37.1 (C-25), 34.2 (C-10), 31.8 (C-1), 31.5 (C-2), 29.6 (C-6), 28.3 (C-16), 25.3 (C-28), 23.0 (C-15), 21.5 (C-11), 21.3 (C-26), 20.9 (C-21), 18.9 (C-27), 25.3 (C-28), 23.0 (C-29); HR-TOFMS (APCI^+^) *m*/*z* 395.3674 [M − H_2_O + H]^+^ (calc. for C_29_H_47_, 395.3672).

Compound **2** was obtained as a colorless needle, mp 263–264 °C (lit [34] 264–266 °C); ${\left[\alpha \right]}_{D}^{25.5}$ −11.5 (c = 0.20, MeOH) (lit [35] ${\left[\alpha \right]}_{D}^{25}$ −24 (c = 0.1, MeOH), [34] [α]_D_ −7.1 (c = 1.61, MeOH)). ^1^H-NMR (300 MHz, CDCl_3_ + DMSO-*d*_6_) δ_H_: 8.56 (br s, 2H, OH), 7.03 (d, *J* = 8.4 Hz, 4H, H-2, 2′, 6 and 6′), 6.69 (dd, *J* = 8.4, 1.8 Hz, 4H, H-3, 3′, 5 and 5′), 4.58 (d, *J* = 4.4 Hz, 2H, H-7 and 7′), 4.07 (br t, *J* = 6.9 Hz, 2H, H-9 and 9′), 3.69 (dd, *J* = 6.3, 1.5 Hz, 2H, H-9 and 9′), 2.98 (br s, 2H, H-8 and 8′); ^13^C-NMR (75 MHz, CDCl_3_ + DMSO-*d*_6_) δ_C_: 156.5 (C-4 and 4′), 131.3 (C-1 and 1′), 127.1 (C-2, 2′, 6 and 6′), 115.2 (C-3, 3′, 5 and 5′), 85.5 (C-7 and 7′), 71.2 (C-9 and 9′), 53.7(C-8 and 8′); HR-TOFMS (ESI^−^) m/z 297.1121 [M − H]^−^ (calc. for C_18_H_17_O_4_, 297.1132). X-ray crystallographic data for Compound **2**: C_18_H_17_O_4_, M = 297.32, Monoclinic, Space Group *P21/c*, *a* = 5.8554(7) Å, *b* = 7.3656(9) Å, *c* = 16.959(2) Å, *α* = *γ* = 90°, *β* = 91.421(4)°, *V* = 731.20(15) Å3, *Z* = 2, *D*calcd = 1.350 Mg/m^3^, crystal size 0.32 × 0.11 × 0.07 mm^3^, F(000) = 314, 25268 reflection collected, 1817 independent reflections (*R*_int_ = 0.0359), *R*_1_ = 0.0549 [*I* > 2*σ*(*I*)], *wR*_2_ = 0.1521 [*I* > 2*σ*(*I*)], *R*_1_ = 0.0617 (all data), *wR*_2_ = 0.1574 (all data), goodness of fit = 1.153.

A single crystal of **2** was pasted on MiTeGen micromounts using paratone oil. X-ray scattering data were collected using a BRUKER D8 QUEST CMOS PHOTON II (Bruker Switzerland AG, Fallanden, Switzerland) operating at T = 296(2) K. The collection of data used ω and ϕ scans and Mo-Kα radiation (λ = 0.71073 Å). The program APEX3 calculates, runs, and images the total number, and unit cell indexing was refined using SAINT [36]. Data reduction was performed using SAINT, and SADABS was used for absorption correction. The ShelXT structure solution program was used and combined dual-space recycling methods [37] and Patterson were used to solve the structure. The structure was refined by least squares using ShelXL [38] and the OLEX2 interface [39]. In the final refinement cycles, all nonhydrogen atoms were cultured anisotropically. All hydrogen atoms were settled in different Fourier maps but were refined using a rigid model (C−H = 0.93–0.98 Å and O−H = 0.82 Å). CCDC-2102982, containing supplementary crystallographic data, can be used free of charge from the Cambridge Crystallographic Data Centre via www.ccdc.cam.ac.uk/data_request/cif (accessed on 13 August 2021).

### 2.5. The Effect of Isolated Compounds on PC-3 Cell Viability

The activities of the pure compounds after isolation and identification were investigated for their antiproliferative effects on PC-3 cells compared to Vero cells. The MTT assay was performed using α-spinasterol at 0–100 µM and ligballinol at 0–500 µM for 72 h, while the vehicle control was treated with both 0.5% DMSO and 1% EtOH. As illustrated in Figure 9, an inhibitory effect was clearly seen for ligballinol. The antiproliferative activity was demonstrated in PC-3 cells with an IC_50_ of 230 μM, but not in normal Vero cells. These results suggest that ligballinol may be an active compound responsible for the anticancer activity of the stem extract. Ligballinol has a furanoid lignan structure and has demonstrated cytotoxic activities against human colorectal cancer cells (HT-29) with an IC_50_ value of 45.5 µM [35]. α-Spinasterol could induce DNA fragmentation and upregulate p53 in breast cancer cells (MCF-7) and ovarian cancer cells (SKOV-3) [40,41]. However, α-spinasterol did not cause cytotoxicity in PC-3 cells, which correlated with our work [42].

## 3. Materials and Methods

### 3.1. General Procedures

^1^H- and ^13^C-NMR spectra were recorded on a Bruker Avance 300 FT-NMR spectrometer (Bruker Switzerland AG, Fallanden, Switzerland) operating at 300 MHz (^1^H) and 75 MHz (^13^C) using CDCl_3_ as the solvent. Specific optical rotations were measured using a Jasco-1020 polarimeter (Jasco Corporation, Tokyo, Japan). Column chromatography and TLC were carried out using Merck silica gel 60 (>230 mesh) and precoated silica gel 60 F_254_ plates, respectively. Spots on the TLC were visualized under UV light (245 or 365 nm) and by spraying with anisaldehyde–H_2_SO_4_ reagent followed by heating at 80 °C until maximum color formation occurred. Only analytical grade chemicals were used: JC-1 reagent (Biotium, Biotium, Inc., Fremont, CA, USA), Hoechst 33342 (Invitrogen, Thermo Fisher Scientific, Inc., Waltham, MA, USA), MTT or 3-(4,5-dimethylthiazol-2-yl)-2,5-diphenyltetrazolium bromide (Sigma-Aldrich, St. Louis, MO, USA), fetal bovine serum (FBS), and DMEM (Gibco, Thermo Fisher Scientific, Inc., Waltham, MA, USA). DMSO and PI (propidium iodide) reagents were purchased from Merck KGaA, St. Louis, MO, USA. 

### 3.2. Plant Materials

MC plants were locally cultivated in a community in Nakhon Pathom Province, Thailand. Leaves, seeds, stems, and roots of the adult plant were collected in September 2016. A voucher has been deposited under number SC001 at the Laboratory of Natural Product Research Unit, Department of Chemistry, Faculty of Science, Srinakharinwirot University, Thailand.

#### 3.2.1. Preparation of MC Extracts

Each plant part was thoroughly washed with tap water, cut into small pieces, and sun dried. The leaves, seeds, stems, and roots of the MC (each 100 g) were individually macerated in MeOH (500 mL) 3 times for two weeks each at room temperature. After that, the methanol extract was filtered (Whatman no. 1 paper), combined, and evaporated to dryness using a rotary evaporator at 40 °C. The obtained dried MeOH extracts (leaves 3.15 g, seeds 6.25 g, stems 0.77 g, and roots 0.8 g) were stored at −20 °C prior to cytotoxic screening. All parts were dissolved in dimethyl sulfoxide (DMSO) by using a water-bath sonicator until completely dissolved.

#### 3.2.2. Phytochemical Isolation of the Stem Extract

For further phytochemical isolation of the stem extract that exhibited the highest cytotoxicity screening, dried stems (16 kg) were collected and then extracted with MeOH at room temperature to yield a brownish residue (320 g). Next, the dried MeOH stem extract was suspended in water and successively partitioned with ethyl acetate (EtOAc) and *n*-butanol (*n*-BuOH), and the combined solution of each extract was evaporated under reduced pressure to yield the EtOAc (greenish sticky mass, 61 g) and *n*-BuOH (brownish sticky mass, 17 g) extracts. Based on the yield of the two fractions, the EtOAc soluble fraction was considered as the main fraction and was subjected to further isolation. The EtOAc fraction (56 g) was subjected to column chromatography (CC) using a gradient system of hexane–EtOAc (98:2 to 0:100) as the eluent to produce 6 main fractions (E1–E6) based on the thin-layer chromatography (TLC) investigations. An oily fraction E1 (26.4 g, 0.16% *w*/*w* based on the dry plant weight) was then evaluated for its fatty acid content. Subfractions E2–E5 displayed similar TLC results, and a colorless solid of α-spinasterol (**1**, 979 mg) was precipitated from fraction E2 (4.3 g). Fraction E6 (12.5 g) was subjected to repeated CC eluting with a gradient of EtOAc–CH_2_Cl_2_ (98:2 to 0:100) to obtain 10 subfractions (E.6.1–E.6.10). Subfraction E.6.5 (4.5 g) was further purified by CC using the same eluent to provide 8 subfractions (E6.5.1–E6.5.8). Ligballinol (**2**, colorless needle, 27 mg) was successfully obtained from subfraction E.6.5.4 (2.8 g).

#### 3.2.3. Analysis of Fatty Acid Composition

##### Preparation of Fatty Acid Methyl Esters

A solution of methanolic sodium hydroxide (9:1, 5 mL) was added to an oil sample (30 mg) and the mixture was heated under reflux for 90 min. After cooling, deionized water (10 mL) was added and the mixture was shaken for a few seconds. Unsaponified fatty acids were discarded by extraction with hexane (3 × 5 mL). The layer below was collected and adjusted to pH 3 (by 6 N HCl), followed by extraction with hexane (3 × 5 mL). The organic layer was removed under reduced pressure. The obtained fatty acid was further mixed with a solution of 2% H_2_SO_4_ in methanol, and the mixture was heated under reflux at 80 °C for 90 min. After cooling to ambient temperature, water (0.5 mL) was added, followed by extraction with hexane (3 × 5 mL). The methyl ester was evaporated to dryness, weighed and solubilized in hexane (1 mL) before injection into the gas chromatograph.

*GC FID-analysis*. Gas chromatography was performed with a Shimadzu version 3, model GC-17A, equipped with a flame ionization detector (FID). FAME analyses were achieved on a Phenomenex BPX70 capillary column (30 m, 0.25 mm i.d.). The initial oven temperature for the analysis of the injection port and FID detector was programmed at 60 °C for 0.5 s and heated to 200 °C for 4 min. The injector and detector temperatures were 250 °C. Helium was used as the carrier gas at a flow rate of 1.25 mL/min, and the split-less time was 1 min for both instruments. The experiment was analyzed by Shimadzu GC solution software V. 2.10, and the results were compared with known standards of fatty acids.

### 3.3. Cell Lines and Cell Culture

The human androgen-independent prostate adenocarcinoma grade IV (PC-3) cell line (ATCC^®^ CRL-1435™), human non-small-cell lung cancer (NSCLC) (A549) cell line (ATCC^®^ CCL-185™), human breast adenocarcinoma (MDA-MB-231) cell line (ATCC^®^ HTB-26™), and normal African green monkey kidney (Vero) cell line (ATCC^®^ CCL-81™) were purchased from ATCC (American Type Culture Collection). All cancer cell lines were grown in high glucose Dulbecco’s modified Eagle’s medium (DMEM) supplemented with 10% fetal bovine serum (FBS), 0.1 mg/mL streptomycin, and 100 units/mL penicillin. The Vero cell line, a representative normal cell line, was maintained in Eagle’s minimum essential medium (EMEM) containing 10% FBS and 1% penicillin/streptomycin at 37 °C in an atmosphere of 5% CO_2_.

### 3.4. Cell Viability Assay

The assay was performed using the standard MTT method. Cells were seeded in a 96-well plate at a density of 1 × 10^4^ cells/well and incubated for 24 h. Next, the cells were treated with various concentrations of the compound tested, including MC crude extracts (0–5 mg/mL), α-spinasterol (0–100 µM), or ligballinol (0–500 µM), while the concentrations of both dimethyl sulfoxide (DMSO; 0.5%) and ethanol (EtOH; 1%) were used as the vehicle control. After treatment, the medium was removed, and MTT solution (0.5 mg/mL) was added to each well. The culture plate was incubated for 4 h at 37 °C in the dark. Next, the supernatant was discarded and DMSO was added to each well to dissolve the formazan precipitate. The optical density (OD) at a wavelength of 570 nm was measured with a multimode microplate reader (Synergy; BioTek Instruments, Inc., Santa Clara, CA, USA) The inhibitory concentration at 50% (IC_50_) was calculated with GraphPad Prism 5.03 software (GraphPad Software, Inc., San Diego, CA, USA).

### 3.5. Nuclei Staining

Apoptotic nuclei were investigated by using Hoechst 33342 staining. Briefly, the cells were seeded in a 6-well plate at a density of 5 × 10^5^ cells/well and incubated for 24 h. The next day, when the cells reached >80% confluence, the media was replaced with 2 mL of fresh complete medium containing 0, 0.3, 0.6, 0.9, and 1.2 mg/mL stem extract for 24 h. Next, the cells were stained with Hoechst 33342 (10 μM) for 30 min at 37 °C in the dark. The characteristics of the cells with fragmented and condensed nuclei were recorded under a fluorescence microscope (IX73; Olympus, Tokyo, Japan) using a magnification of ×20 for 24 h.

### 3.6. Sub-G1 Apoptosis Assay

Sub-G1 peak analysis was performed using a flow cytometer. Cells were seeded in a 6-well plate at a density of 5 × 10^5^ cells per well and incubated for 24 h at 37 °C in a 5% CO_2_ atmosphere. Next, the cells were treated with various concentrations (0, 0.6, 0.9, and 1.2 mg/mL) of stem extract. After incubation for 24 h, the cells were harvested and fixed with 70% ice-cold ethanol at 4 °C for at least 1 h. Subsequently, the cells were washed with ice-cold PBS, followed by incubation in 20 μg/mL RNase A (Sigma, St. Louis, MO, USA) and 50 µg/mL PI (Merck KGaA, St. Louis, MO, USA) with 1.5% Triton X-100 for 30 min at 37 °C in the dark. The cells were analyzed with a flow cytometer (Guava EasyCyte and GuavaSoft software version 3.3 (Merck KGaA, St. Louis, MO, USA)). The data are depicted as a histogram and the percentage of sub-G1 peaks display the apoptotic populations.

### 3.7. Measurement of Mitochondrial Membrane Potential (MMP)

The MMP was measured using JC-1 fluorescent staining. To perform the experiment, the cells were treated with various concentrations (0, 0.6, 0.9, and 1.2 mg/mL) of stem extract, and 0.5% DMSO was used for the control group. Next, the cells were collected by trypsinization and centrifuged at 500× *g* for 5 min at 4 °C, followed by PBS washing. After that, the cells were incubated with JC-1 dye for 15 min, centrifuged at 500× *g* for 5 min, resuspended in PBS, and analyzed with flow cytometry (Guava EasyCyte and Guava Mitopoteintial software (Merck KGaA)).

### 3.8. Western Blot Analysis

Cells were seeded in 6-well plates at a density of 5 × 10^5^ cells per well in DMEM containing 10% FBS and cultured for 24 h. Next, the cells were treated with various concentrations (0, 0.3, 0.6, 0.9, and 1.2 mg/mL) of stem extract. After incubation, the cells were lysed with RIPA buffer (1 M Tris-HCl pH 7.4, 5 M NaCl, 20% NP-40, 10% sodium deoxycholate, 20% SDS, and a protease inhibitor cocktail) on ice for 30 min and centrifuged at 14,000 rpm at 4 °C for 20 min. The concentration of protein was analyzed using a Bradford protein assay. Following protein quantification, 12% SDS-polyacrylamide gel electrophoresis (SDS–PAGE) separated 20–30 µg of the total protein extracts and was then transferred to PVDF membranes and blocked with 5% skim milk in TBS-T buffer (5% *w*/*v* skim milk, 0.1% Tween-20 and 1× Tris-buffered saline solution) for 1 h at room temperature, then incubated with primary antibodies against caspase-3 (1:1000; product no. 9662), Mcl-1 (1:1000; product no. 5453T), Bcl-xl (1:1000; product no. 2764T), GAPDH (1:1000; product no. 2118S; all from Cell Signaling Technology, Inc., Danvers, MA, USA), and Noxa (1:1000; product no. 14766S) at 4 °C overnight. The membranes were washed 3 times and then incubated with anti-rabbit IgG antibodies (1:5000; product no. 7074P2) or horseradish peroxidase-conjugated anti-mouse antibodies ((1:5000; product no. 7076P2); both from Cell Signaling Technology, Inc.) for 1 h. The membranes were visualized by enhanced chemiluminescence using ECL plus™ Western blotting detection reagents (Bio-Rad Laboratories). The intensities of the protein bands were quantified by ImageJ software [Java 1.8.0_112 (64_bit)].

### 3.9. Statistics 

All results are presented as the mean ± standard error (SEM, of each group = 3). Statistical analysis was performed using GraphPad Prism version 5.03 (GraphPad Software, Inc.). Statistical comparisons with paired observations were made by a one-way analysis of variance followed by Dunnett’s test. *p* < 0.05 was considered to indicate a statistically significant difference.

## 4. Conclusions

In the present study, the MC extract from leaves, seeds, stems, and roots demonstrated cytotoxic effects in several types of cancer cells, including lung, cervical, breast, and prostate cancer cells. However, the most effective extract was from the stem part, which not only exhibited a strong effect on cancer cells but also showed less toxicity on normal cells. In addition, the stem extract clearly indicated apoptosis induction in prostate cancer cells. Additionally, the stem extract showed an interesting ability to reduce the expression of anti-apoptotic proteins (Bcl-xL and Mcl-1) and increase the expression of pro-apoptotic proteins (caspase-3 and Noxa) in prostate cancer cells. The isolated compound ligballinol may be responsible for this function; however, additional investigations are necessary.

## Figures and Tables

**Figure 1 molecules-27-01313-f001:**
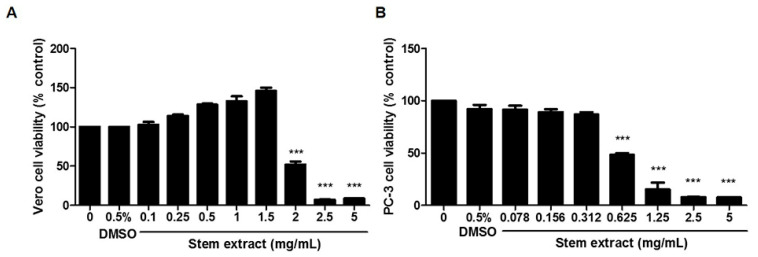
Dose-dependent inhibitory effects of an MC stem extract on the viability of (**A**) Vero and (**B**) PC-3 cells. Cells were treated with various concentrations (0–5 mg/mL) for 24 h. Data are presented as the mean of triplicate values ± SEM. *** indicates *p* < 0.001.

**Figure 2 molecules-27-01313-f002:**
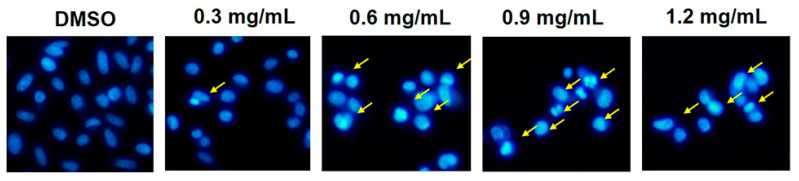
Fragmentation and nuclear condensation were observed by Hoechst 33342 staining. PC-3 cells were treated with stem extract (0, 0.3, 0.6, 0.9, and 1.2 mg/mL) for 24 h. Treated cells exhibited morphological changes in the nuclei typical of apoptosis. Photographs were taken under a fluorescence microscope (20×). Arrows represent apoptotic cells.

**Figure 3 molecules-27-01313-f003:**
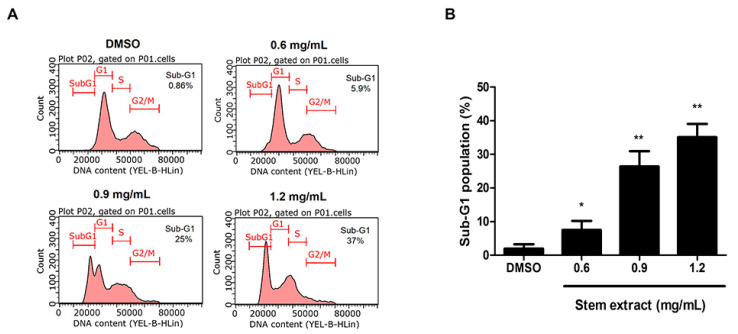
Sub-G1 peak analysis of PC-3 cells treated with stem extract. The flow cytometry analyzed PC-3 cells that were treated with stem extract at 0, 0.6, 0.9, and 1.2 mg/mL for 24 h. (**A**) Histograms demonstrate the number of cells (y-axis) vs. the DNA content (x-axis). The data shown are representative of three experiments with similar findings. (**B**) The quantitative values of the sub-G1 population. The significant differences of the treated cells from the untreated control group are indicated by * *p* < 0.05 and ** *p* < 0.01.

**Figure 4 molecules-27-01313-f004:**
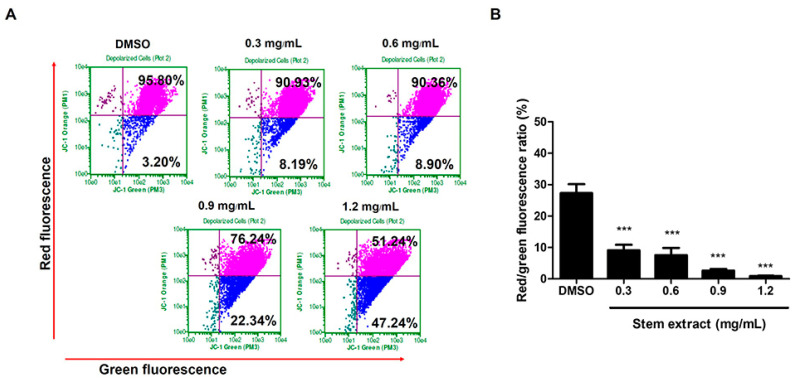
Effects of stem extract on the mitochondrial membrane potential (ΔΨm) in PC-3 cells (**A**) Cells were treated with stem extract at 0, 0.3, 0.6, 0.9, and 1.2 mg/mL for 24 h. Next, all samples were stained with JC-1 and analyzed by flow cytometry. The upper right quadrant indicates polarized mitochondria (red fluorescence), while the lower right quadrant illustrates membrane depolarization (green fluorescence). (**B**) Quantification of the red/green fluorescence ratio indicates low membrane potential following increasing concentrations of the stem extract. *** indicates *p* < 0.0001 compared to the control.

**Figure 5 molecules-27-01313-f005:**
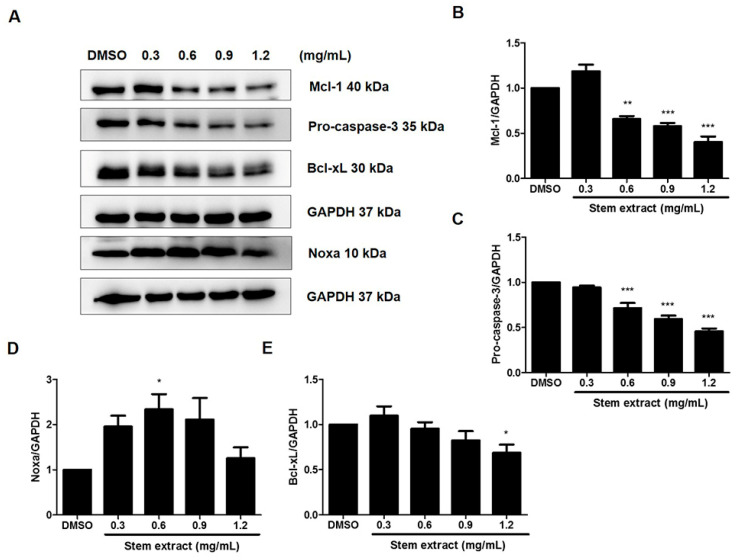
Effects of the stem extract on the expression of apoptosis-related proteins in PC-3 cells. Cells were treated with stem extract at 0, 0.3, 0.6, 0.9, and 1.2 mg/mL, and then Western blot analysis was performed with all samples. (**A**) Representative protein expression of McL-1, pro-caspase-3 Bcl-xL, and Noxa revealed by Western blot analysis. (**B**–**E**) The expression levels of each protein were quantified and normalized to GAPDH. The results are expressed as means ± SEM. * indicates *p* < 0.05, ** indicates *p* < 0.01, and *** indicates *p* < 0.001: a significant difference versus the control.

**Figure 6 molecules-27-01313-f006:**
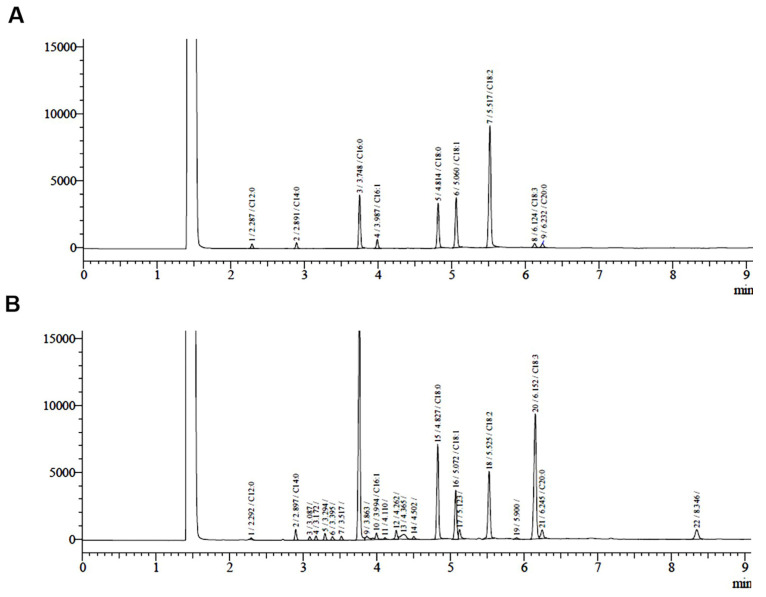
Chromatograms of the fatty acid content of the oil fraction obtained by the GC-FID analysis. (**A**) Standard chromatogram of 9 fatty acids that are known as lauric acid (C12:0), myristic acid (C14:0), palmitic acid (C16:0), palmitoleic acid (C16:1), stearic acid (C18:0), oleic acid (C18:1), linoleic acid (C18:2), α-linolenic acid (C18:3), and arachidic acid (C20:0). (**B**) Identification chromatogram of fatty acids from the sample.

**Figure 7 molecules-27-01313-f007:**
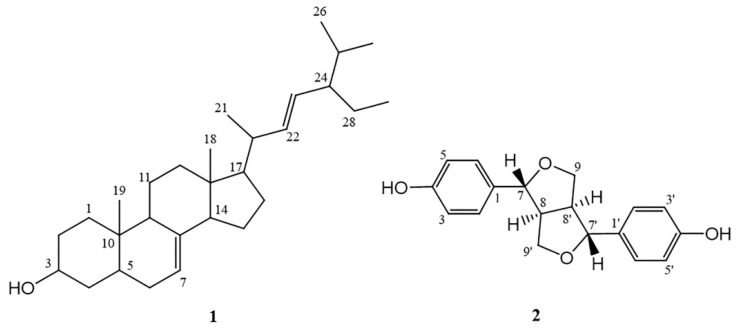
Chemical structures of α-spinasterol (**1**) and ligballinol (**2**).

**Figure 8 molecules-27-01313-f008:**
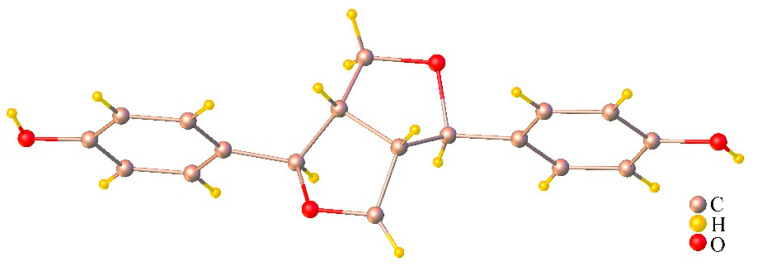
ORTEP plot of the X-ray crystal structure for ligballinol.

**Figure 9 molecules-27-01313-f009:**
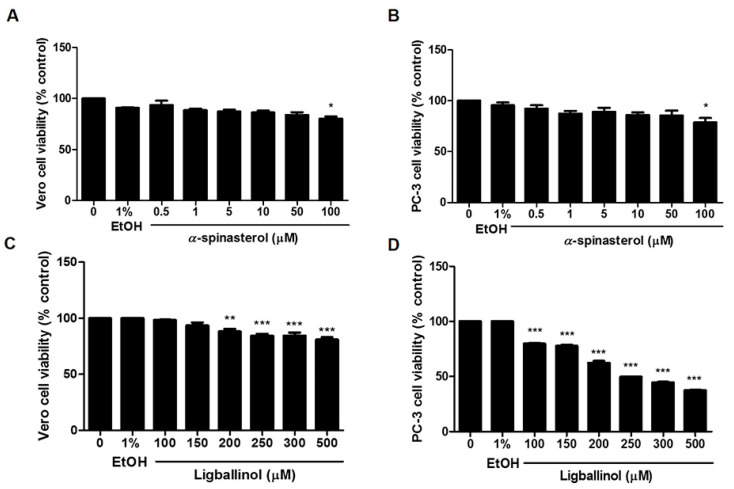
The antiproliferative effect of α-spinasterol and ligballinol on PC-3 and Vero cells. Cells were treated with various concentrations of (**A**,**B**) α-spinasterol and (**C**,**D**) ligballinol, and MTT assays were conducted. The results are expressed as means ± SEM. * indicates *p* < 0.05, ** indicates *p* < 0.01, and *** indicates *p* < 0.001: a significant difference versus control.

**Table 1 molecules-27-01313-t001:** The antiproliferative effect of MC extracts were determined by MTT assays. All cells were treated with or without the different parts (leaves, seeds, stems, and roots) of extracts (0–5 mg/mL) for 24 h. Control cells were exposed to vehicle or 0.5% DMSO. The data are offered as the mean of triplicate values ± SEM.

Cell Lines	IC_50_ Values (mg/mL)			
	Leaves	Seeds	Stems	Roots
A549	1.98 ± 0.74	1.30 ± 0.96	0.44 ± 0.87	0.74 ± 3.14
HeLa	0.53 ± 0.37	2.09 ± 1.41	0.42 ± 1.07	0.48 ± 0.38
MDA-MB-231	2.10 ± 0.79	2.26 ± 1.18	0.62 ± 0.74	0.77 ± 3.72
PC-3	4.25 ± 0.66	4.65 ± 0.36	0.62 ± 0.74	0.73 ± 2.27

**Table 2 molecules-27-01313-t002:** The fatty acids identified from the MC stem extracts by GC/FID analysis.

Peak	RT	Fatty Acid Names	Mean ± SEM
1	2.292	Lauric acid (C12:0)	0.18 ± 0.01
3	2.897	Myristic acid (C14:0)	1.54 ± 0.02
8	3.761	Palmitic acid (C16:0)	35.07 ± 0.19
10	3.994	Palmitoleic acid (C16:1)	1.34 ± 0.02
15	4.827	Stearic acid (C18:0)	14.09 ± 0.05
16	5.072	Oleic acid (C18:1)	7.30 ± 0.02
18	5.525	Linoleic acid (C18:2)	11.85 ± 0.07
20	6.125	α-Linolenic acid (C18:3)	27.0 ± 0.09
21	6.245	Arachidic acid (C20:0)	1.63 ± 0.16

## Data Availability

All the data of the present study are presented in the manuscript.

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
