# Peer review of "Stem Extract from *Momordica cochinchinensis* Induces Apoptosis in Chemoresistant Human Prostate Cancer Cells (PC-3)"

_molecules, 2022, doi:10.3390/molecules27041313_

Round 1

Reviewer 1 Report

The authors have satisfactorily updated the manuscript as requested earlier.

Reviewer 2 Report

Revised manuscript Stem extract from Momordica cochinchinensis induces apoptosis in chemoresistant human prostate cancer cells (PC-3) is now recommended to be accepted in Molecules in present form.

Reviewer 3 Report

Final I can suggest to accept the Manuscript, although I am still not sure about its reliable scientific soundness. Technicaly and formaly it is OK.

This manuscript is a resubmission of an earlier submission. The following is a list of the peer review reports and author responses from that submission.

Round 1

Reviewer 1 Report

Authors have undoubtedly contributed to prostate cancer research by identifying a new source of drug for the treatment. It is a well-written manuscript. I have a few minor comments.

  1. All species names should be in italics.
  2. Figure 7 and 8 seems to be in poor resolution.
  3. The abstract may include a sentence regarding the background for choosing the plant.

Author Response

The authors have already edited as the reviewer's suggestion. The details are indicated as file attached.

Reviewer 2 Report

This manuscript describes „Stem extract from Momordica cochinchinensis induces apoptosis in chemoresistant human prostate cancer cells (PC-3)” and has been submitted as an article.

The main goal of the study was the isolation of stem extracts and new, novel components of the extract from this really important species, to determine their structures and examine their antitumor (better to say „antiproliferative” or selective cytotoxic”) activities.

With both parts of the aims, I have serious problems.

With the novelty of the isolated compounds: all of them are known in the literature, eg. ligballinol and spinasterol have been already isolated from this plant and was examined on different cancer celll lines (though not on prostata cancer cells) and was found practically ineffective (Geng, Yi-man; Zhao, Lian-mei; Zhu, Xiu-li; Ren, Feng-zhi; Han, Li-na; Gao, Hai-xiang; Shan, Bao-en: Zhongcaoyao (2013), 44(14), 1951-1956). Similarly just recently (in press article) anticancer activity of different extract from Momordica cochinchinensis was studied and found practically uneffective (Anticancer Activity and Molecular Mechanism of Momordica cochinchinensis Seed Extract in Chronic Myeloid Leukemia Cells: Zhengdong Ai,Chong et al. l: Nutrition and cancer, Published online: 15 Dec 2021, doi.org/10.1080/01635581.2021.2014904). As I see from the manuscript, neither the extracts nor the two isolated compounds have no remarkable( (IC50 > 100 micromol practically inactive).  Generally over 30 mikroM concentration, there is no reason to talk antiproliferative activity at all.

The only novelty that they succesfully analized the component of the stem extract (the seed has been already published elsewhere), so although formally the manuscript is well-written and in the content I have found only a few typing mistakes and some other small problems (e.g. the enantiopurity of the isolated (-)ligballion: the optical rotation is much weaker than that in the literature which makes question aboóut the optical purity, m.p. of ligballinol) the main problem is the missing of novelty and, I suppose, the scientific soundness of these result.

Because of the abovementioned inextricable problems of the MS I can not do any, just  strongly suggest to REJECT the manuscript.

Author Response

The authors have already edited and responded to reviewer's suggestion. 

Point 1: This manuscript describes “Stem extract from Momordica cochinchinensis induces apoptosis in chemoresistant human prostate cancer cells (PC-3)” and has been submitted as an article.    The main goal of the study was the isolation of stem extracts and new, novel components of the extract from this really important species, to determine their structures and examine their antitumor (better to say „antiproliferative” or selective cytotoxic”) activities.

Responses 1: Thank you for your valuable comments. We agree that “antiproliferative” is more suitable than antitumor. So, we have edited as shown in Line 29, 84, 86, 90, 94, 97, 112, 281, and 292

Point 2: With both parts of the aims, I have serious problems. With the novelty of the isolated compounds: all of them are known in the literature, eg. ligballinol and spinasterol have been already isolated from this plant 

Response 2: We agree with reviewer that the compound a-spinasterol and ligballinol have already isolated from this plant and screened for the anticancer activity. However, as we have mentioned that both a-spinasterol and ligballinol have never been reported in prostate cancer cells, so we thought that this is a valuable data since PC3 cell is an aggressive cancer type that usually resistance to chemotherapy.

Point 3: the isolated compounds: all of them are known in the literature, eg. ligballinol and spinasterol have been already isolated from this plant and was examined on different cancer cell lines (though not on prostate cancer cells) and was found practically ineffective (Geng, Yi-man; Zhao, Lian-mei; Zhu, Xiu-li; Ren, Feng-zhi; Han, Li-na; Gao, Hai-xiang; Shan, Bao-en: Zhongcaoyao (2013), 44(14), 1951-1956).

Response 3: we quite disagree with this comment. Because we have looked through this article (it is written in Chinese, and we used the translation tool) and found that ligballinol could inhibit the proliferation of B16 melanoma cells at 40% at 8 mg/ml.

Point4: ) anticancer activity of different extract from Momordica cochinchinensis was studied and found practically uneffective 

Response4: we quite disagree. From the literature review, several works showed that extracts from MC showed a cytotoxic effect in different types of cancer cells, we have already mentioned as shown in references No.9-12, 14-17.

Point 5: anticancer activity of different extract from Momordica cochinchinensis was studied and found practically uneffective (Anticancer Activity and Molecular Mechanism of Momordica cochinchinensis Seed Extract in Chronic Myeloid Leukemia Cells: Zhengdong Ai,Chong et al. l: Nutrition and cancer, Published online: 15 Dec 2021, 

Response 5: we have found that this article mentioned the effect of seed extract from MC that could inhibit the growth of KBM5 (Chronic myeloid leukemia cells) at 34.36 μg/ ml. So, it effectively inhibits the growth of cancer cells.   

Point 6: As I see from the manuscript, neither the extracts nor the two isolated compounds have no remarkable( (IC50 > 100 micromol practically inactive).  Generally over 30 mikroM concentration, there is no reason to talk antiproliferative activity at all.

Response 6: we agree that the value is quite high. However, the concentration range of more than 30 mM has been reported in other natural compounds to inhibit the growth of PC3 cells. For example, myricetin, an abundant flavonoid found in the bark and leaves of bayberry, shows multiple promising anti-tumor functions in various cancers. The IC50values of myricetin in PC3, and DU145 were at 47.6 µM and 55.3 µM, respectively. (The Natural Compound Myricetin Effectively Represses the Malignant Progression of Prostate Cancer by Inhibiting PIM1 and Disrupting the PIM1/CXCR4 Interaction; Ye C.et al; Cell Physiol Biochem 2018;48:1230–1244).

Point 7: some other small problems (e.g. the enantiopurity of the isolated (-)ligballion: the optical rotation is much weaker than that in the literature which makes question about the optical purity, m.p. of ligballinol) 

Response 7: Thank you for reviewer’s kind comment.  The discussion on both isolated compounds including their physical data has been amended as marked in yellow.  Please note that the negative specific rotation value of compound 2 seems have variations.

Line  215

……. .”Fraction E2-5 and E6.5.4 were collected and proved to be α-spinasterol (1) and (‒) ligballinol (2), respectively, by spectroscopic examination, mainly 1H and 13C NMR and MS data and X-ray crystallography (Figure 7‒8) (Supplementary material 1‒6) as well as by comparison of their physicochemical and spectral data with published values. Compound 2 exhibited a negative specific rotation [α]D25.5 ‒11.5 (c = 0.20, MeOH), and its X-ray crystal structure supported the stereochemical assignment, as shown (Figure 8). Therefore, Compound 2 was deduced to be (‒) 4,4'-((1R,3aS,4R,6aS)-tetrahydro-1H,3H-furo[3,4-c]furan-1,4-diyl)diphenol or (‒) ligballinol. Compound 2 also has been isolated from the MC seeds [a].

Line 233

Compound 1 was obtained as a white amorphous powder, mp 168-169 oC (lit [b] 169 oC) …………………………………………………………………………………………………………………………

HR-TOFMS (APCI+) m/z 395.3674 [M - H2O + H]+ (calc. for C29H47, 395.3672).

Line 244

Compound 2 was obtained as a colorless needle, mp 263-264 oC (lit [c] 264-266 oC); [α]D25.5 ‒11.5 (c = 0.20, MeOH) (lit [31] [α]D25 ‒24 (c = 0.1, MeOH), [c] [α]D ‒7.1 (c = 1.61, MeOH))…………………………………

…………………………………………………………………………………………………………………………

(C-9 and 9′), 53.7 (C-8 and 8′); HR-TOFMS (ESI-) m/z 297.1121 [M - H]- (calc. for C18H17O4, 297.1132).

Reviewer 3 Report

Manuscript entitled Stem extract from Momordica cochinchinensis induces apoptosis in chemoresistant human prostate cancer cells (PC-3) is well prepared and is recommended to be published in Molecules after minor revision.

Please consider following suggestions and corrections:

Lines 16, 22, 28, 30 and 61: Momordica cochinchinensis Latin names in italic.

Line 86: MDA-MB-231  with hyphen

Line 101: in Figure 1   without dot after Figure. Same in lines 123, 126, 140, 275

Table 1: Spaces are left on each side of the ±: e.g. 1.98 ± 0.74. Same in Table 2.

Line 143: mL instead of ml

Line 158: Leave a space before and after mathematical operators that function as verbs or conjunctions; that is, they have numbers on both sides or a symbol for a variable on one side and a number on the other: *P < 0.05. Same in lines 165, 197, 198, 287, 432

Line 213: (1) – brackets not in bold. Same in lines 214, 267, 446, 447 and 448

Lines 217, 240, 241: D in subscript

Lines 219-220: IUPAC name with descriptors in italic: 4,4'-((1R,3aS,4R,6aS)-tetrahydro-1H,3H-furo[3,4-c]furan-1,4-diyl)diphenol

Lines 232, 233, 234, 242, 243: J in italic

Lines 233, 234: CH3 with number in subscript

Lines 239 and 246: Can you provide HR-MS data for compounds 1 and 2?

Line 241: δH

Line 266: Please provide Figure 7 in a higher resolution.

Line 280: What does the number 32 mean?

Line 281: [36,37].

Line 296: H2SO4

Line 332: CH2Cl2

Lines 340, 342, 345: (3 × 5 mL)          × instead of x

Line 370: Spaces are left on each side of the ×: 1 × 104   Same in lines 383, 391, 409

Line 391: CO2

Lines 404, 405: 500 × g    g in italic

Line 413: 14,000 g

Author Response

The authors have edited all suggestions from the reviewer as shown in the manuscript file attached.  

Round 2

Reviewer 2 Report

All the correctable request were done, but I keep my opinion about the relatively low novelty and weak scientific soundness of the manuscript (certainly it can not be improved in this stage). In case IC50>30 micromol we practicaly not give any data... But formally all the possible changes are well-done, so even my suggestion is still "reject", but if the editor choice is acceptable, I will not complain.